# Comprehensive Constitutional Genetic and Epigenetic Characterization of Lynch-Like Individuals

**DOI:** 10.3390/cancers12071799

**Published:** 2020-07-05

**Authors:** Estela Dámaso, Maribel González-Acosta, Gardenia Vargas-Parra, Matilde Navarro, Judith Balmaña, Teresa Ramon y Cajal, Noemí Tuset, Bryony A. Thompson, Fátima Marín, Anna Fernández, Carolina Gómez, Àngela Velasco, Ares Solanes, Sílvia Iglesias, Gisela Urgel, Consol López, Jesús del Valle, Olga Campos, Maria Santacana, Xavier Matias-Guiu, Conxi Lázaro, Laura Valle, Joan Brunet, Marta Pineda, Gabriel Capellá

**Affiliations:** 1Hereditary Cancer Program, Catalan Institute of Oncology, Insititut d’Investigació Biomèdica de Bellvitge (IDIBELL), ONCOBELL Program, Avinguda de la Gran Via de l’Hospitalet 199-203, 08908 L’Hospitalet de Llobregat, Barcelona, Spain; edamaso.riquelme@gmail.com (E.D.); migonzalez@iconcologia.net (M.G.-A.); gvargas@idibell.cat (G.V.-P.); mnavarrogarcia@iconcologia.net (M.N.); fmarin@idibell.cat (F.M.); afrobles@iconcologia.net (A.F.); cgomez@iconcologia.net (C.G.); asolanes@iconcologia.net (A.S.); siglesias@iconcologia.net (S.I.); jdelvalle@iconcologia.net (J.d.V.); ocampos@iconcologia.net (O.C.); conxi.lazaro@gmail.com (C.L.); lvalle@iconcologia.net (L.V.); Jbrunet@iconcologia.net (J.B.); 2Centro de Investigación Biomédica en Red de Cáncer (CIBERONC), 28029 Madrid, Spain; avelasco@iconcologia.net (À.V.); fjmatiasguiu.lleida.ics@gencat.cat (X.M.-G.); 3High Risk and Cancer Prevention Group, Vall d’Hebron Institute of Oncology (VHIO), Carrer de Natzaret 115-117, 08035 Barcelona, Spain; jbalmana@vhio.net; 4Medical Oncology Department, Hospital de Santa Creu i Sant Pau, Carrer de Sant Quintí 89, 08041 Barcelona, Spain; tramon@santpau.cat (T.R.y.C.); clopezsa@santpau.cat (C.L.); 5Genetic Counseling Unit, Hospital Arnau de Vilanova, Avinguda Alcalde Rovira Roure 80, 25198 Lleida, Spain; ntusetda@gmail.com (N.T.); giselaurgell@gmail.com (G.U.); 6Faculty of Medicine, Dentistry and Health Sciences, University of Melbourne, Building 181 Grattan St, VIC 3010 Melbourne, Australia; Bryony.Thompson@mh.org.au; 7Hereditary Cancer Program, Catalan Institute of Oncology, Institut d’Investigació Biomèdica de Girona (IDIBGI), Carrer del Dr. Castany s/n, 17190 Salt, Girona, Spain; 8Pathology Department, Hospital Arnau de Vilanova, Institut de Recerca Biomèdica de Lleida (IRB Lleida), Avinguda Alcalde Rovira Roure 80, 25198 Lleida, Spain; msantacana@irblleida.cat; 9Pathology Department, Bellvitge University Hospital, Insititut d’Investigació Biomèdica de Bellvitge (IDIBELL), Carrer de la Feixa Llarga s/n, 08907 L’Hospitalet de Llobregat, Barcelona, Spain; 10Department of Medical Sciences, School of Medicine, University of Girona, Carrer Emili Grahit 77, 17003 Girona, Spain

**Keywords:** Lynch syndrome, Lynch-like syndrome, variant of unknown significance, epimutation, mismatch repair, methylation, cancer genes panel, next generation sequencing

## Abstract

The causal mechanism for cancer predisposition in Lynch-like syndrome (LLS) remains unknown. Our aim was to elucidate the constitutional basis of mismatch repair (MMR) deficiency in LLS patients throughout a comprehensive (epi)genetic analysis. One hundred and fifteen LLS patients harboring MMR-deficient tumors and no germline MMR mutations were included. Mutational analysis of 26 colorectal cancer (CRC)-associated genes was performed. Pathogenicity of MMR variants was assessed by splicing and multifactorial likelihood analyses. Genome-wide methylome analysis was performed by the Infinium Human Methylation 450K Bead Chip. The multigene panel analysis revealed the presence of two MMR gene truncating mutations not previously found. Of a total of 15 additional MMR variants identified, five -present in 6 unrelated individuals- were reclassified as pathogenic. In addition, 13 predicted deleterious variants in other CRC-predisposing genes were found in 12 probands. Methylome analysis detected one constitutional *MLH1* epimutation, but no additional differentially methylated regions were identified in LLS compared to LS patients or cancer-free individuals. In conclusion, the use of an ad-hoc designed gene panel combined with pathogenicity assessment of variants allowed the identification of deleterious MMR mutations as well as new LLS candidate causal genes. Constitutional epimutations in non-LS-associated genes are not responsible for LLS.

## 1. Introduction

Lynch syndrome (LS) is a hereditary cancer predisposition syndrome that increases the risk for colorectal and endometrial cancer as well as other tumors [1]. It is mainly caused by pathogenic germline (epi)genetic alterations in the mismatch repair (MMR) genes *MLH1, MSH2, MSH6* and *PMS2* [1]. Inactivation of the MMR wildtype allele is needed for tumor development, leading to an MMR-deficient phenotype typically characterized by loss of expression of MMR proteins and microsatellite instability. In MMR-deficient sporadic tumors, MLH1 loss of expression is mainly due to somatic *MLH1* promoter methylation [2].

Even in the absence of somatic *MLH1* promoter methylation, no MMR germline pathogenic variants are identified as a causal mechanism in approximately 55% of patients showing MMR-deficiency in tumors; constituting the so called Lynch-like syndrome (LLS) [3]. LLS is considered a heterogeneous group showing intermediate risk of colorectal cancer (CRC) between LS and sporadic cancer [4,5]. Thus, the identification of causal mechanisms is crucial for guiding individualized surveillance strategies for LLS patients and their relatives.

Constitutional (germline) MMR cryptic mutations (usually associated to rearrangements or regulatory regions), somatic mosaicism and variants of unknown significance occur in a proportion of LLS cases [6,7,8,9,10,11,12]. Furthermore, double somatic hits in MMR genes have been detected in a variable proportion (30–82%) of LLS [9,10,13,14,15,16,17]. However, even in the presence of double somatic MMR hits, an inherited predisposition to cancer -unrelated to MMR genes- cannot be totally excluded [9,18]. Biallelic *MUTYH* mutations, commonly associated with attenuated familial adenomatous polyposis, have been detected in 1 to 3% of LLS patients [19,20,21]. Likewise, germline mutations in proofreading polymerases can lead to MMR-deficiency [22]. Recently other genes are emerging as LLS candidate causal genes, such as *MCM9, FAN1*, *BUB1*, *SETD2, EXO1, RFC1* and *RPA1* [10,23,24,25,26].

Constitutional epigenetic alterations in *MLH1* and *MSH2* are occasionally responsible for the MMR-deficient phenotype in LS patients [27]. Similarly, constitutional epigenetic alterations have been rarely described in other cancer genes such as *BRCA1* and *RAD51C* in ovarian and breast cancer [28], *KILLIN* in Cowden syndrome [29] or *DAPK* in chronic lymphocytic leukemia [30]. In contrast, the role of constitutional methylation in LLS has not been yet explored.

The aim of the current study is to elucidate the constitutional basis of MMR deficiency in a cohort of 115 LLS cases throughout a comprehensive genetic and epigenetic characterization. The obtained results contribute to the understanding of LLS by ruling out the presence of constitutional methylation events as a common cause for LLS as well as highlighting the relevance of performing comprehensive genetic analyses in these patients.

## 2. Materials and Methods

### 2.1. Patients

A total of 115 Caucasian Lynch-like syndrome patients harboring MMR deficient tumors MMR loss of expression and/or microsatellite instability (MSI) were included (Appendix A). Twenty-three of them were reported in a previous publication [10]. The immunohistochemistry (IHC) pattern of MMR protein expression was as follows: 57 MLH1/PMS2 loss, 27 MSH2/MSH6 loss, 12 MSH6 loss, five PMS2 loss and 14 MMR conserved expression but MSI. In the 57 tumors showing loss of MLH1/PMS2 protein expression the presence of somatic *MLH1* promoter hypermethylation and/or *BRAF* V600E were excluded, except for three cases (7, 9 and 78) that had wildtype *BRAF* and non-informative tumor *MLH1* promoter methylation results.

Based on the IHC MMR expression pattern, the corresponding MMR genes were sequenced. Cases in whom no pathogenic variants in MMR genes had been identified were included in this study (Appendix A). Of note, nine patients initially classified as LLS were excluded from this cohort due to the previous identification of germline biallelic *MUTYH* and *MSH2* pathogenic mutations [10,19,31]. Concerning clinical criteria fulfillment, 83 patients met Revised Bethesda guidelines (72.2%) and 11 the Amsterdam criteria (9.6%) for hereditary nonpolyposis CRC (Appendix A). The remaining 21 (5.4%) were referred to the Genetic Counseling Unit because of histological features suggestive of MMR-deficiency and loss of MMR protein expression.

In addition to LLS patients, 61 LS cases harboring MMR genetic mutations, 12 constitutional *MLH1* epimutation carriers and 41 healthy controls were included as controls for genome-wide methylome analysis [32] (Appendix A).

All patients were assessed at the Cancer Genetic Counseling Units of the Catalan Institute of Oncology, Santa Creu i Sant Pau, Arnau de Vilanova and Vall d’Hebron hospitals from 1998 to 2012. Informed consent was obtained from all individuals enrolled and internal Ethics Committee approved this study (code PR225/11).

### 2.2. Samples

Isolation of genomic DNA from blood of all included patients was performed using FlexiGene DNA kit (Qiagen, Hilden, Germany) or Wizard Genomic DNA Purification Kit (Promega, Madison, WI, USA) according to the manufacturer’s instructions.

FFPE blocks of normal colorectal mucosa and CRC tissue were obtained when available. For each FFPE specimen, 10-20 x 10-μm sections were cut from a single representative block per case, using macrodissection with a scalpel if needed to enrich for tumor cells. After deparaffinization using the Qiagen Deparaffinization Solution (Qiagen), DNA was isolated using the QIAmp DNA FFPE Tissue Kit (Qiagen) according to manufacturer’s instructions.

DNA quality was tested using a NanoDrop ND 1000 Spectrophotometer (ThermoFisher Scientific, Waltham, MA, USA), electrophoresis in agarose gel and a Qubit Fluorometer (Invitrogen, Carlsbad, CA, USA) using the dsDNA BR Assay (Invitrogen, Carlsbad, CA, USA).

### 2.3. Mismatch Repair Genes Mutational Analysis

#### 2.3.1. Mutational Analysis of Coding Regions of MMR Genes

According to the IHC pattern in tumors, mutation analysis of candidate MMR genes (*MLH1* NM_000249.3, NG_007109.2; *MSH2*, NM_000251.2, NG_007110.1; *MSH6*, NM_000179.2, NG_007111.1; *PMS2* NM_000535.6, NG_008466.1) was initially performed on blood DNA by PCR amplification of exonic regions and exon–intron boundaries or Single Strand Conformation Polymorphism (SSCP), followed by Sanger sequencing. Primers and conditions are available upon request. Genomic rearrangements in the MMR genes were analyzed by multiplex ligation dependent probe amplification (MLPA) using SALSA-*MLH1*/*MSH2* P003-B1, SALSA-*MLH1*/*MSH2* P248-B1, *MSH6* P072 and/or PMS2 P008-C1 kits (MRC-Holland), according to manufacturer’s indications. Screening of gross rearrangements in MSH2-deficient cases was complemented by using the 2 available MLPA kits for *MSH2* gene analysis and by screening the recurrent *MSH2* inversion in exons 1–7 [11]. Annotation of variants was done following the Human Genome Variation Society recommendations. Variants were classified according to Insight classification guidelines [33].

#### 2.3.2. Direct Sequencing of MMR Promoter Regions and 3′UTR of the EPCAM Gene

The regions encompassing 662 bases upstream of the transcriptional start site (TSS) of *MSH2*, 915bp of *MSH6* TSS, 1469bp of *MLH1* TSS and 429bp of the *EPCAM* 3′UTR were amplified by PCR using Megamix-Double (Microzone Ltd., Haywards Heath, UK) and sequenced using the BigDye Terminator v.3.1 Sequencing Kit (Applied Biosystems, Foster City, CA, USA) (Appendix A; conditions available upon request). Sequences were analyzed on an ABI Prism 3100 Genetic Analyzer (Applied Biosystems).

### 2.4. Targeted Next Generation Sequencing

Sixty-two LLS patients with strong individual and/or familial cancer history (Amsterdam or Bethesda 1, 2, 4 or 5 criteria) were analyzed using a NGS custom panel of 26 CRC associated genes, previously used for the characterization of MSH2/MSH6–deficient cases [10]. Agilent SureDesign web-based application (Agilent Technologies, Santa Clara, CA, USA) was used to design DNA capture probes of 509 target regions, including the coding exons plus 10 flanking bases of 26 genes associated to CRC, as well as their promoter regions (comprising 650 bases upstream their TSS), as previously reported [10]. Agilent SureCall application was used to trim, align and call variants. Variant filtering was performed based on Phred quality ≥30, alternative allele ratio ≥0.05, read depth ≥38x in PBL samples. Identified variants were then filtered against common single-nucleotide polymorphisms (MAF > 1% according to ExAC and ESP databases) as well as class 1 and class 2 MMR variants according to InSight database. Predicted pathogenic germline rare variants were further confirmed by Sanger sequencing using independent DNA samples. Primers and conditions are detailed in Appendix A.

### 2.5. Pathogenicity Assessment of Genetic Variants

#### 2.5.1. Variant Frequency and Cosegregation Analysis

Global population frequency of the identified variants was retrieved from the Exome Aggregation Consortium (ExAC; http://exac.broadinstitute.org/) and NHLBI Exome Sequencing Project (ESP; http://evs.gs.washington.edu/EVS) databases. Identified variants were also screened in DNA samples from family relatives by Sanger sequencing when available.

#### 2.5.2. In Silico Prediction of the Functional Impact

Alamut Visual v2.9.0 software (Interactive Biosoftware, Rouen, France) was used for in silico predictions. The potential effects of variants on splicing were evaluated by using SSF, MaxEnt, NN SPLICE and Gene Splicer. At the protein level the impact of variants was analyzed using the in silico algorithms PolyPhen-2, SIFT, Align GVGD and Mutation taster. Also, PROVEAN was used for in-frame indel variants. PROMO 3.0 software was used to predict any changes in transcription factor binding between wildtype alleles and promoter variants. Only human transcription factors were considered and 5% was selected as maximum matrix dissimilarity rate.

#### 2.5.3. Multifactorial Likelihood Analysis

For MMR variants, posterior probability of pathogenicity was calculated by multifactorial likelihood analysis as previously described [34,35] based on estimated prior probabilities of pathogenicity and likelihood ratios (LR) for segregation and tumor characteristics. Variants were classified according to the five class IARC scheme based on the calculated posterior probability.

#### 2.5.4. mRNA Splicing Analysis and Allele Specific Expression Analysis

Available lymphocytes from variant carriers were cultured with and without puromycin after one week of culture with PB-MAX medium. Total RNA was extracted using Trizol Reagent. One microgram of RNA was retrotranscribed using iScript cDNA synthesis kit (Bio-Rad, Hercules, CA, USA). cDNA amplification of exon containing the variants and at least two exons up and downstream the main one was performed using specific primers provided in Appendix A. Sequencing was performed using the BigDye Terminator v.3.1 Sequencing Kit (Applied Biosystems, Foster City, CA, USA). Mutation Surveyor (SoftGenetics, State College, PA, USA) was used for sequence visualization.

For allelic expression analyses, regions containing heterozygous variants were selected. The relative levels of both alleles were determined in genomic DNA and cDNA by single-nucleotide primer extension (SNuPE) as previously described [36] (primers provided in Appendix A). Allele-specific expression (ASE) was calculated by dividing the ratio of variant/wildtype allele in cDNA by the ratio of variant/wildtype allele in gDNA. Experiments were performed in quadruplicate. ASE values of 1.0 indicate equal levels of expression from both alleles. ASE values lower than 1.0 indicate reduced expression from one allele.

### 2.6. Tumor Analysis

Whole exome sequencing of FFPE DNA extracted from the tumor of patient 53 -carrier of a germline variant in the exonuclease domain of POLE- and of his blood, was carried out in a Hi-Seq2000 (Illumina) with a coverage >100x, after library preparation using the Agilent Sure Select Human All Exon v5 kit. Sequence alignment was carried out with BWA and variant calling with MuTect. Variants identified in the patient’s blood DNA were eliminated for the analysis of somatic mutations in the tumor. Variants present in at least 10% of the reads were considered for subsequent analyses. The contribution of the COSMIC mutational signatures [37] to the tumor was calculated with *deconstructSigs* [38].

MSH3 expression and elevated microsatellite instability at selected tetranucleotide repeats (EMAST) were evaluated in the normal and tumor samples from case 74, harboring two *MSH3* variants. Immunohistochemistry of MSH3 protein was performed using anti-MSH3 antibody at dilution 1:150 (Novus Biologicals, Centennial, CO, USA). The reaction was visualized with the EnVision^TM^ FLEX Detection Kit (Agilent Technologies-DAKO, Santa Clara, CA, USA) following standard protocols. For EMAST analysis, six previously reported tetranucleotide repeat markers were analyzed [39]. Primers and conditions are listed in Appendix A. The amplification products were run on an ABI Prism 3130 DNA sequencer and analyzed using GeneMapper v4.0 (Applied Biosystems). EMAST was considered when two or more of the analyzed markers displayed instability.

### 2.7. Genome-Wide Methylation Profiling

Blood DNA samples from LLS patients and controls, as well as available FFPE colorectal normal/tumor DNA, were included in the genome wide methylation profiling analysis using Infinium Human Methylation 450K Beadchip (Appendix A), also including the LLS cases previously reported [10].

Array data processing and data analysis were performed as previously described [32]. Blood DNA with an A260/A280 ratio between 1.7–2.0 were considered suitable for hybridization. DNAs from FFPE samples were analyzed by qPCR using Infinium FFPE QC (Illumina, Cambridge, UK) in order to determine their suitability for FFPE restoration. All samples showing ΔCt values lower than 5 were restored using the Infinium HD FFPE Restore kit (Illumina), following the manufacturer’s instructions. A total of 1000 ng blood DNA and 500 ng FFPE DNA were bisulfite converted using the EZ DNA Methylation™ Kit (Zymo Research, Irvine, CA, USA), according to the manufacturer’s instructions. To determine the efficiency of the bisulfite conversion, a predetermined genomic region was evaluated by Sanger sequencing in one methylated and one unmethylated control of each bisulfite conversion batch. Genome wide methylation profiling was performed using the Infinium Human Methylation 450K Bead Chip (Illumina), which interrogates the methylation status of 485.764 CpG sites across the genome. For internal quality control, in vitro methylated and unmethylated DNAs were included in each batch. After hybridization, sample scanning was performed using the HiScan platform (Illumina), which has a laser scanner with two colours (532 nm/660 nm). The relative intensity of each dye was analyzed using the GenomeStudio software (Methylation Module). For each analyzed CpG site, a β-value was obtained depending on the florescence intensity. Β measures took values between 0 (unmethylated) and 1 (fully methylated). The analysis of batch effects was performed using RnBeads software (Max-Planck-Institute Informatik, Saarbrücken, Alemania). Group comparisons and statistical analysis -based on differentially methylated CpG sites, CpG islands, promoters, genes and tiling- were performed using RnBeads software (Max-Planck-Institute Informatik). CpG methylation was visualized using the Integrative Genome Viewer (Broad Institute, Cambridge, MA, USA). GRCh37/hg19 was used as the reference genome (date of release: February 2009). Only positions that reached an FDR *p*-value < 0.05 when comparisons are done between groups > 10 samples were considered.

### 2.8. Availability of Data

A schematic workflow of the study design and the obtained results are presented in Figure A1.

The datasets generated and analyzed during the current study are available in the GEO repository:

Lynch-like series: https://www.ncbi.nlm.nih.gov/geo/query/acc.cgi?acc=GSE128068.

Control series: https://www.ncbi.nlm.nih.gov/geo/query/acc.cgi?acc=GSE107353.

## 3. Results

### 3.1. Reassessment of Germline Genetic Variants in the MMR Genes

The presence of missed MMR genetic alterations was reassessed in blood samples from 42 LLS patients with strong individual and/or familial cancer history by means of a NGS custom panel of CRC-associated genes, previously used in the analysis of 23 MSH2-deficient LLS cases from the same series [10] (Table 1). By using this approach two *bona fide* previously not identified germline pathogenic MMR variants were found in two cases fulfilling the Amsterdam criteria (cases 33 and 92).

Case 33 was a male who suffered from two CRCs at age of 40 and 46 (Figure 1). Immunohistochemical (IHC) staining displayed loss of MLH1 protein expression in his first tumor, being non-informative the second one. Previous SSCP analysis was negative whereas NGS analysis identified a pathogenic *MLH1* mutation, c.676C>T (p.Arg226*).

Case 92 was a woman who developed endometrial cancer at age 48 (Figure 1). Her tumor displayed MSI with conserved MMR protein expression. No mutation was identified by Sanger sequencing in either *MLH1* or *MSH2*. The panel allowed the identification of a truncating mutation in *MSH6,* c.2219T>A (p.Leu740*). In addition to the nine MMR variants of unknown significance identified in 10 LSS individuals in previous analyses (cases 35, 39, 58, 63, 67, 70, 72, 73, 75 and 77), four additional variants (*MSH6* c.2092C>G, *MSH6* c.3150_3161dup, *PMS2* c.1320A>G and MSH2 c.2802G>A) were detected in four additional cases (cases 5, 82, 85 and 98, respectively) (Table 1).

This re-analysis was complemented with the sequencing of the promoter regions of the four MMR genes, which identified an *MLH1* promoter variant (c.-574T>C, rs558088820, MAF <0.0001) in case 13 (Table 1). This variant was predicted to interfere with YY1 transcription factor binding, which directs histone deacetylases and histone acetyltransferases to the promoter in order to activate or repress its activity [40]. Regarding rearrangements, the presence of the germline recurrent inversion of exons 1–7 in MSH2-deficient cases [11] was evaluated with negative results (Table 1). In contrast, MLPA reanalysis using the P248 kit (MRC-Holland) revealed the presence of an *MSH2* exon 8 duplication in case 57 (Table 1 and Figure 1).

### 3.2. Pathogenicity Assessment of MMR Variants

In all, 15 MMR VUS were identified in 16 probands (Table 2): seven in *MSH6,* five in *MSH2*, two in *MLH1* and one in *PMS2*. mRNA splicing evaluation and stability analyses were possible for the *MSH6* variants c.1153_1155del (p.Arg385del), c.1618_1620del (p.Leu540del) and c.3150_3161dup (p.Val1051_Ile1054dup). An aberrant transcript at low proportion was identified in the two c.1618_1620del carriers (cases 70 and 75) corresponding to a partial out-of-frame deletion of exon 4 (r.1607_3172del, p.Ser536_Asp1058delinsAsn), which coexisted with the full-length transcript (r.1618_1620del, p.Leu540del) (Appendix A). This agrees with a partial allelic imbalance detected at the c.1618 position (Table 2). The remaining two variants analyzed had no apparent effect on mRNA splicing and stability (Table 2). Clinico-pathological data from the same families were used in multifactorial likelihood analyses. Information from *MSH6* c.1153_1155del and c.3150_3161dup carriers later identified in our centers was also included in the multifactorial calculations (AF1-3; Appendix A). For the three *MSH6* variants, posterior probability of pathogenicity resulted >0.98, classifying c.1618_1620del and c.3150_3161dup as pathogenic, and c.1153_1155del as probably pathogenic (Table 3). In addition, *MSH6* c.3226C>T (p.Arg1076Cys) variant, initially classified as VUS, was reclassified as probably pathogenic (class 4) because of its co-occurrence *in trans* with *MSH6* pathogenic mutations in patients with constitutional MMR deficiency and loss of MSH6 expression in normal cells [41,42].

No effect on splicing and transcript stability was detected in lymphocytes from the carrier of *MSH2* c.1787A>G (p.Asn596Ser) variant, as previously reported [43] (Table 2). In case 57, splicing analysis confirmed the presence of an aberrant transcript containing the exon 8 duplication (r.1277_1387dup), predicted to generate a frameshift protein (p.Val463Glufs*11), thus allowing to classify the variant as pathogenic (Appendix A).

The functional impact of *MLH1* promoter c.-574T>C variant on *MLH1* transcription could not be assessed due to the absence of coding heterozygous *MLH1* variants, being therefore classified as VUS. Likewise, the other nine variants identified in MMR genes remained as VUS due to insufficient evidence, although in silico predictions suggested neutrality for four of them (*MSH2* c.1787A>G, c.2045G>C and c.2802G>A and *PMS2* c.1320A>G) (Table 2 and Appendix A).

### 3.3. Identification of Variants in Other CRC-Predisposing Genes

The multigene panel analysis allowed the identification of rare germline variants in other CRC-predisposing genes in 32 LLS cases (32/42, 76.2%) (Appendix A). Thirteen of them were variants predicted as pathogenic by in silico tools, identified in well-known CRC predisposing genes such as *APC* and *MUTYH*, as well as variants in newly emerging cancer predisposing genes such as *MSH3* and *FAN1* (Table 4 and Appendix A). Among them, four variants were identified in the *MSH3* gene (Table 4), two of them coexisting in *cis* in the same patient (case 74; Appendix A). One of these two variants, c.2732T>G (p.Leu911Trp) affects a highly conserved residue along MutS proteins, and the other one, c.685T>C (p.Tyr229His), is located next to the DNA recognition domain of the protein and affects a highly conserved residue [44]. While immunohistochemical staining showed conserved MSH3 nuclear expression in normal and tumor tissue from case 74, tetranucleotide repeats analysis displayed instability in two out of six microsatellites, indicating EMAST (Appendix A).

The *FAN1* c.149T>G (p.Met50Arg) variant was found in heterozygosity in case 39, diagnosed with CRC at 49 years of age. This variant, localized at the ubiquitin-binding domain, was previously associated to pancreatic cancer predisposition [47]. Functional assays demonstrated that c.149T>G variant affects *FAN1* nuclease activity, impeding the repair of chromosome abnormalities when forks stall after hydroxyurea and mitomycin treatment [48]. Conversely, homozygous carriers of this *FAN1* variant have been reported in the Genome Aggregation Database (GnomAD).

*POLE* c.898A>G (p.Ile300Val) variant, located within the exonuclease domain of the polymerase, was identified in patient 53, diagnosed with CRC at age 51 and two synchronous CRC at age 81. Tumor WES revealed a major contribution of COSMIC mutational signature 6 (56.2%), associated with MMR deficiency, and complete absence of the POLE-associated COSMIC mutational signature 10, or signature 14, identified in tumors with concurrent POLE mutation and MMR deficiency [37] (Appendix A). The evidence gathered indicates lack of causal association of the *POLE* c.898A>G with the patient’s CRC, and supports a benign nature of the variant, as suggested by the in silico tools.

*EXO1* c.2212-1G>A was identified in case 58, diagnosed with CRC at age 58 and 61. The splice-site variant causes an in-frame deletion of six amino acids in the MSH2 interaction domain (Table 4). The absence of family history prevented cosegregation analysis.

No rare (population MAF < 0.01) germline variants were identified in *BUB1B, CHEK2, PTEN, STK11* or *TP53* genes (Appendix A).

### 3.4. Constitutional Epigenetic Alterations in MMR Genes

Methylome analysis was firstly used to evaluate the existence of constitutional epigenetic alterations in the MMR genes. Blood DNA from case 7 displayed *MLH1* promoter hypermethylation that was further validated in blood using MS-MLPA (mean methylation in the *MLH1* C/D regions 48%; data not shown). The *MLH1* epimutation carrier developed a *BRAF* wildtype CRC at age 42 (Figure 2A). Blood methylation pattern matched in extension with the 1.6 Kb differentially methylated region (DMR) previously described in constitutional epimutation carriers [32] (Figure 2B). The constitutional epimutation was also detected in normal colorectal mucosa of the carrier (Figure 2C). No other cases with MMR promoter hypermethylation were found.

### 3.5. Global Epigenetic Characterization of Lynch-Like Cases

Constitutional genome-wide epigenetic characterization of LLS cases was carried out with the aim of assessing the contribution of constitutional epimutations in other non-LS genes to LLS. No differentially methylated (DM) CpG islands were evidenced when LLS blood samples were compared to LS or healthy individuals (Appendix A). The *EPM2AIP1-MLH1* CpG island was the only DM region identified in blood when the LLS group was compared to *MLH1* constitutional epimutations (Appendix A). The subsequent analysis of individual CpG sites identified several DM sites in the genome (Appendix A). Among them, only a single CpG located within *KHDC1* gene showed methylation differences higher than 20% in *MLH1*-deficient LLS cases in comparison to constitutional *MLH1* epimutations. However, this CpG site, located in a boundary between a non-methylated and a fully methylated region, evidenced high dispersion within groups (Appendix A). No constitutional epigenetic aberrations were evidenced in the LLS group when methylome data was reanalyzed after excluding LS variant carriers and carriers of predicted pathogenic variants in CRC predisposing genes.

Next, we investigated the presence of tissue-specific epigenetic alterations in normal colorectal mucosa. Similar to the results obtained in blood samples, no DM CpG islands or CpG sites were identified in LLS when compared to LS or healthy control samples (Appendix A). No further differences were observed when analyzing the colorectal tumors from LLS and LS patients (Appendix A). Methylome analysis of DM CpG islands in paired normal-tumor colonic samples from LLS individuals resulted in the identification of a high number of DM CpG islands (n = 4380), most of them (n = 3076) also identified as DM in normal-tumor samples from LS individuals (Figure 3), pointing to similar tumor methylation patterns in both groups. As previously reported [49], strong hypermethylation of CpG islands and moderate hypomethylation of CpG sites within body genes was observed in the tumors from both groups.

## 4. Discussion

Individuals with MMR deficient tumors and no identified germline MMR mutations account for more than a half of the cases being assessed at genetic counseling units because of LS suspicion. They encompass a heterogeneous group of patients that may benefit from further stratification after comprehensive (epi)genetic characterization [50]. By combining the use of variant pathogenicity assessment with ad-hoc designed panel and a global epigenetic characterization, we reclassified 9 of 115 cases as LS, one secondary to a constitutional epimutation. These results, together with the 5 cases from the same series reclassified in a previous work from our group [10], yielded a 12% (14/120) reclassification rate. Also, predicted deleterious variants in other CRC predisposing genes were found, which might explain, at most, an additional 11% of LLS cases. Except for the *MLH1* constitutional epimutation, no other clinically relevant differentially methylated regions were identified in LLS after a genome-wide methylome analysis.

In the present work, a customized NGS panel for the analysis of 26 CRC-associated genes allowed us to identify two previously missed bona fide MMR pathogenic variants in two families fulfilling the Amsterdam criteria. Fifteen additional MMR variants (nine identified by previous Sanger sequencing and six in the current MMR gene re-analysis) were also found in 16 individuals. RNA analyses in combination with multifactorial likelihood calculations resulted in the classification of five of them as (probably) pathogenic mutations. These results highlight the benefit of applying quantitative and qualitative analyses for variant interpretation and classification. Of note, four out of the 17 identified MMR variants (including pathogenic mutations and VUS) were not found in the candidate MMR gene according to the IHC pattern (cases 5, 82, 92 and 98), two of them finally classified as disease causing in the family (cases 82 and 92). These observations highlight the benefit of multiplex MMR gene panel testing in the presence of discordant IHC results.

Copy number variant (CNV) reanalysis using an updated MLPA test identified an *MSH2* exon 8 duplication in an additional case fulfilling Amsterdam criteria. These results further reinforce the notion that reanalysis of MMR genes using updated testing strategies should be considered in former LLS cases with strong individual and/or familial cancer history. While our NGS panel was not designed for CNV identification, recent advances in bioinformatic analysis have allowed the robust identification of rearrangements in other cancer gene panels, making it closer to the routine use of NGS for CNV identification [51].

The different molecular nature of the pathogenic MMR variants identified in our work, including single nucleotide and splicing variants, rearrangements and epimutations, highlights the need to apply a variety of experimental approaches in the search for the constitutional basis of MMR deficiency. Our comprehensive strategy has proved useful for the elucidation of the underlying molecular basis of a relevant number of suspected LS patients, with the consequent clinical impact for patients and their families.

By using subexome panel analysis previous works reported the identification of candidate genes for LLS [10,23,24]. In our cohort, variants were found in well-known CRC predisposition genes such as *APC* and *MUTYH,* as well as in newly emerging candidate genes for cancer predisposition, such as *MSH3, EXO1* and *FAN1*. Since patients with biallelic mutations in *MUTYH* were previously discarded in our LLS series [19,31], only three heterozygous *MUTYH* carriers were found (current study and [10]). As recommended, the estimated risk for monoallelic *MUTYH* mutation carriers does not support an earlier initiation of colonoscopy screening [50,52].

There are a few reports of germline monoallelic variants in *EXO1* and *MSH3* in LS suspected families, although the clinical significance of these variants has not been yet determined [53,54]. Moreover, *MSH3* variants have been found in combination with variants in LS-associated genes [18,55]. Of note, biallelic *MSH3* mutations have been recently associated with adenomatous polyposis and CRC predisposition [39]. In our cohort, 4 patients were carriers of monoallelic predicted pathogenic variants in *EXO1* or *MSH3* genes, and one *MSH3* carrier case harbored a tumor showing EMAST. These findings suggest the possibility of an oligogenic effect of *MSH3* and *EXO1* variants. Further studies are needed in order to elucidate the role of *MSH3* and *EXO1* in LLS.

Recent reports implicate *FAN1* as a CRC and pancreatic cancer predisposing gene [25,47]. We found a patient carrying the *FAN*1 c.149T>G (p.Met50Arg) variant which was previously associated to functional defects and pancreatic cancer predisposition [47,48]. However, the role of *FAN1* in cancer predisposition is currently a matter of controversy since no significant increase in the burden of *FAN1* mutations are detected in CRC cases versus controls [56,57].

At the epigenetic level, genome-wide methylation profiling was performed in DNA from blood and available colorectal tissue of all probands of our series. Individual methylation analysis of MMR genes allowed the identification of a new case of constitutional *MLH1* epimutation [27,32]. This finding reinforces the need to rule out suggestive *MLH1* epimutation cases by analyzing DNA blood methylation in all early-onset cancer patients, irrespective of family history, where somatic methylation has not been assessed.

In our study, genome-wide methylome analysis ruled out other common constitutional epigenetic alterations associated with LLS individuals. This analysis also discarded the presence of colorectal tissue specific epimutations, as described for *MSH2* epimutations [58]. However, we cannot completely rule out the existence of methylation aberrations in specific groups, considering the diversity of MMR IHC patterns in LLS. Methylome analysis was not able to discriminate between tumors from LLS and LS individuals, in line with the strong homogeneity of the epigenetic and genetic profile of MSI tumors previously reported [59,60].

## 5. Conclusions

In all, germline reassessment of LS suspected cases is useful for the elucidation of the molecular basis of a relevant proportion of LLS cases. Subexome panels of cancer predisposing genes in combination with pathogenicity assessment of variants offered a good yield in reclassification, unmasking the limitations of IHC testing and the difficulty of detecting cryptic MMR mutations. The availability of advanced sequencing technologies will shed light on the molecular classification of LLS at the germline level. When combined with somatic testing these technologies will likely fulfill their anticipated potential.

## Figures and Tables

**Figure 1 cancers-12-01799-f001:**
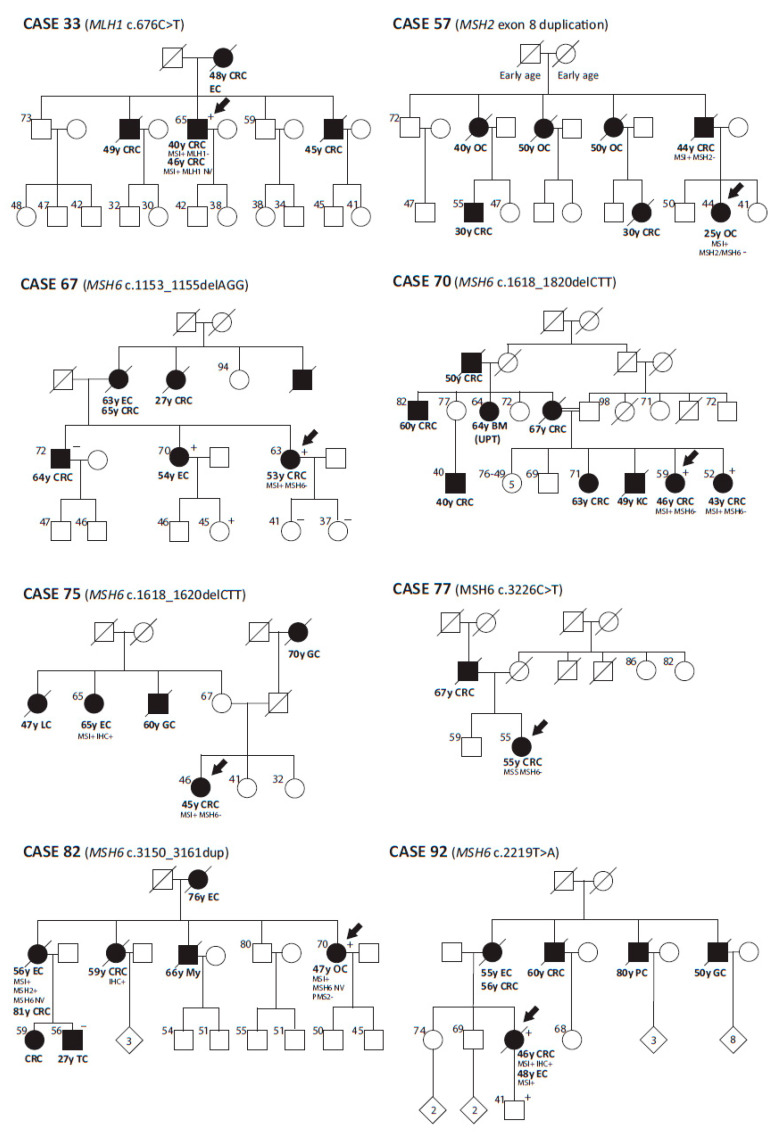
Pedigrees from patients reclassified as Lynch syndrome in the current study. Abbreviations: CRC, colorectal cancer; EC, endometrial cancer, PC, prostate cancer, GC, gastric cancer, OC, ovarian cancer, BM(UTP), brain metastasis from unknown primary tumor, KC, kidney cancer, TC, testis cancer, My, myeloma, MSI+, microsatellite instability, NV, No valuable, +, variant carrier, -, variant non carrier.

**Figure 2 cancers-12-01799-f002:**
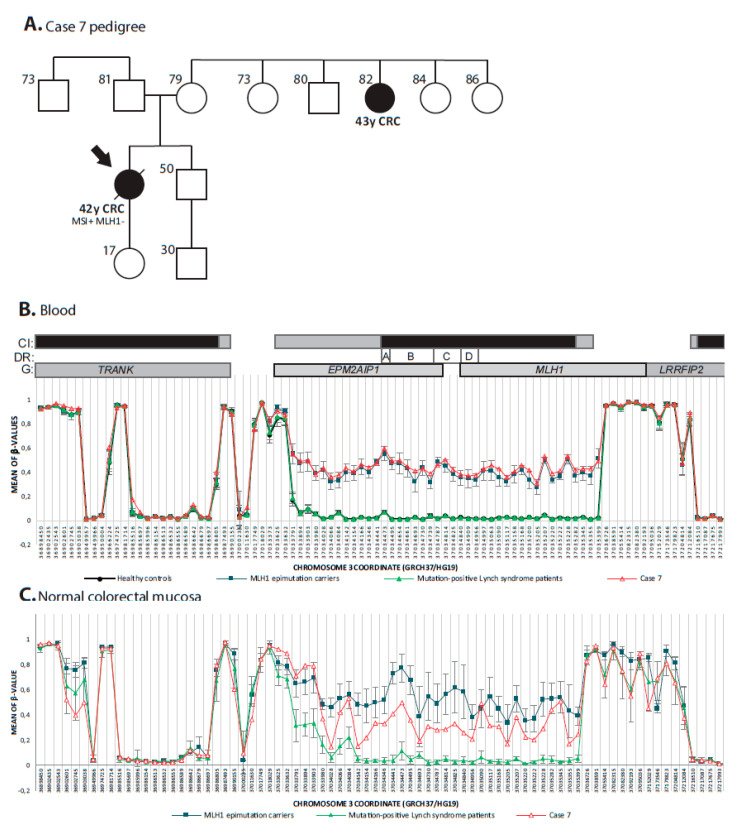
**Identification of a new case of constitutional *MLH1* epimutation**. (**A**) Pedigree of case 7. Representation of mean β-values in blood DNA (**B**) and FFPE normal colorectal mucosa (**C**) from case 7 against *MLH1* epimutation carriers, mutation-positive Lynch syndrome patients and healthy controls at differentially methylated region described for constitutional *MLH1* epimutation carriers. Chromosome coordinates of CpG sites are graphed at axis of abscissa. The location of the CpG sites are not drawn to scale. CpG islands (CI) are represented as black rectangles and their shores in grey. Location of Deng’s promoter regions (DR) are indicated as white rectangles. Genes (G) including displayed CpG sites are represented as grey rectangles. Cytoband divisions (CB) are displayed as grey rectangles. Ensembl GRCh37 was taken as reference for gene coordinates.

**Figure 3 cancers-12-01799-f003:**
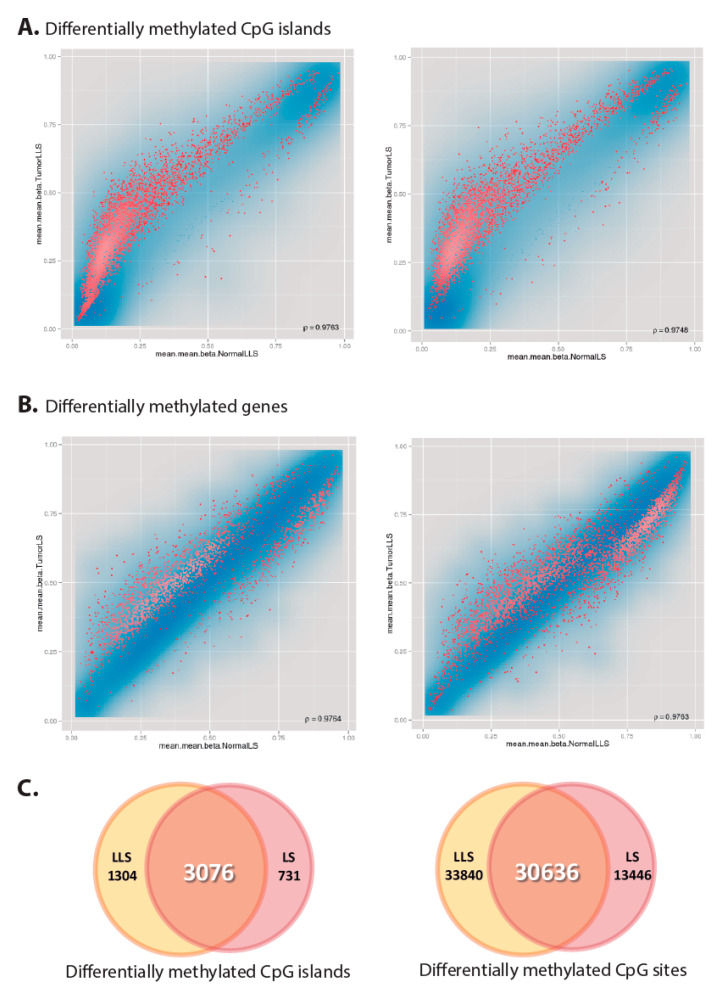
Scatterplot of the normalized mean B-values obtained using the Infinium 450k Human Methylation array to identify differentially methylated CpG islands (**A**) and genes (**B**) in tumors from LLS cases (left) and LS controls (right). The transparency corresponds to point density. One % of the points in the sparsest populated plot regions are drawn explicitly. The colored points represent differentially methylated CpG islands and genes with an FDR adjusted p-values lower than 0,05. (C) Venn diagrams of the differentially methylated CpG islands (left) and CpG sites (right), which shown the overlapping of epigenetic changes during tumorigenesis in LLS cases (yellow) and LS controls (red).

**Table 1 cancers-12-01799-t001:** **Results obtained from the characterization of LLS patients, including cases analyzed by NGS, MMR VUS carriers and epimutants.** (*) Not previously reported MMR variant classified according to InSiGHT variant classification rules. (#) See details in Appendix A. (^) Up to date only *EPCAM* 3′UTR deletions has been associated with CRC predisposition, whereas point mutations cause an autosomal recessive congenital diarrhoea (OMIM:613217) unrelated to CRC.

Case ID	Results from Previous MMR Mutational Analysis by Sanger Sequencing/SSCP (#)		Results from the Analysis of CRC-Associated Genes (This Study)		Pathogenicity Assessment of MMR VUS		Final Case Classification
	Variants in LS-Associated Genes (Insight Classification)		Variants in LS-Associated Genes (Insight Classification)	Predicted Pathogenic Variants in Other Predisposing Genes (ClinVar Classification)	VUS Assessment	Final Variant Classification	
**A. Results obtained in the analysis of 42 samples by using an NGS subexome panel of CRC-associated genes. Only the MMR variants and the predicted pathogenic variants in other genes are shown (see Appendix A).**							
**5**	-		*MSH6* c.2092C>G, p.Gln698Glu (Class 3)	-			LLS (MMR VUS carrier)
**6**	-		-	-			LLS
**7**	-		***MLH1*** **epimutation**	-	Confirmation by MS-MLPA (48%)		**LS (*MLH1* epimutation carrier)**
**8**	-		-	-			LLS
**10**	-		-	*EPCAM^* c.811G>T, p.Val271Phe (not reported)			LLS (VUS carrier)
**13**	-		*MLH1* c.-574T>C, p.? (Class 3*)	-			LLS (MMR VUS carrier)
**28**	-		-	-			LLS
**29**	-		-	*POLD1* c.2275G>A, p.Val759Ile (Class 1,2,3)			LLS (VUS carrier)
**30**	-		-	*APC* c.7936C>G, p.Gln2646Glu (Class 3)			LLS (VUS carrier)
**33**	-		***MLH1* c.676C>T, p.Arg226* (Class 5)**	**-**			**LS**
**39**	*MSH2* c.1787A>G; p.Asn596Ser (Class 3)		*MSH2* c.1787A>G; p.Asn596Ser (Class 3)	*FAN1* c.149T>G, p.Met50Arg (not reported)			LLS (MMR VUS carrier)
**42**	-		-	-			LLS
**44**	-		-	-			LLS (VUS carrier)
**45**	-		-	-			LLS
**48**	-		-	-			LLS
**53**	-		-	-			LLS
**55**	-		-	*PMS1* c.497A>C, p.Lys166Thr (not reported)			LLS (VUS carrier)
**56**	-		-	-			LLS
**57**	-		*MSH2* E8 duplication (Class 3*)	-	Aberrant splicing	***MSH2*** **E8 duplication (Class 5)**	**LS**
**58**	*MSH2* c.2045C>G; p.Thr682Ser (Class 3*)		*MSH2* c.2045C>G; p.Thr682Ser (Class 3*)	*EXO1* c.2212-1G>A (Class 3)			LLS (MMR VUS carrier)
**59**	-		-	*APC* c.1966C>G, p.Leu656Val (Class 3)			LLS (VUS carrier)
**61**	-		-	-			LLS
**62**	-		-	*MSH3* c.2732T>G, p.Leu911Trp (not reported)			LLS (VUS carrier)
**63**	*MSH2* c.2702A>T; p.Glu901Val (Class 3*)		*MSH2* c.2702A>T; p.Glu901Val (Class 3*)	-			LLS (MMR VUS carrier)
**64**	-		-	-			LLS
**65**	-		-	*MUTYH* c.1437_1439delGGA, p.Glu480del (Class 5)			LLS (monoallelic MUTYH carrier)
**66**	-		-	-			LLS
**74**	-		-	*MSH3* c.685T>C, p.Tyr229His (not reported); *MSH3* c.2732T>G, p.Leu911Trp (not reported)	MSH3 conserved expression/EMAST/in cis	*MSH3* c.685T>C, p.Tyr229His (VUS); MSH3 c.2732T>G, p.Leu911Trp (VUS)	LLS (VUS carrier)
**76**	-		-	-			LLS
**78**	-		-	-			LLS
**79**	-		-	-			LLS
**81**	-		-	*BUB1* c.2473C>T, p.Pro825Ser (not reported)			LLS (VUS carrier)
**82**	-		*MSH6* c.3150_3161dup, p.(Val1051_Ile1054dup) (Class 3*)	-	Multifactorial (>0.99)/normal splicing	*MSH6* c.3150_3161dup, p.(Val1051_Ile1054dup) (Class 5)	LS
**85**	-		*PMS2* c.1320A>G, p.Pro440= (Class 3*)	*MSH3* c.3072G>C, p.Gln1024His (not reported)			LLS (MMR VUS carrier)
**87**	-		-	-			LLS
**92**	-		***MSH6* c.2219T>A, p.Leu740* (Class 5)**	-			**LS**
**93**	-		-	-			LLS
**94**	-		-	-			LLS
**95**	-		-	-			LLS
**96**	-		-	*APC* c.7514G>A, p.Arg2505Gln (Class 1,2)			LLS (VUS carrier)
**97**	-		-	-			LLS
**98**	-		*MSH2* c.2802G>A, p.Thr934Thr (Class 3)	-			LLS (MMR VUS carrier)
**B. Results obtained in 7 additional cases harboring MMR variants identified by previous Sanger sequencing**							
**35**	*MLH1* c.25C>T, p.Arg9Trp, (Class 3) *APC* c.1958+3A>G (Class 5) (Borrás et al. 2012)		-	-			FAP (MMR VUS carrier)
**67**	*MSH6* c.1153_1155del, p.Arg385del (Class 3 *)		-		Multifactorial (0.98)/normal splicing	*MSH6* c.1153_1155delAGG p.Arg385del (Class 4*)	LS
**70**	*MSH6* c.1618_1620delCTT; p.Leu540del (Class 3 *)		-	-	Multifactorial (>0.99)/aberrant splicing at low proportion	*MSH6* c.1618_1620delCTT; p.Leu540del (Class 5*)	LS
**72**	*MSH6* c.1450G>A; p.Glu484Lys (Class 3 *)		-	-			LLS (MMR VUS carrier)
**73**	*MSH6* c.3296T>A; p.Ile1099Asn (Class 3 *)		-	-			LLS (MMR VUS carrier)
**75**	*MSH6* c.1618_1620del; p.Leu540del (Class 3 *)		-	-	Multifactorial (>0.99)/aberrant splicing at low proportion	*MSH6* c.1618_1620delCTT; p.Leu540del (Class 5*)	LS
**77**	*MSH6* c.3226C>T, p.Arg1076Cys (Class 3)		-	-	Insight variant classification revision	*MSH6* c.3226C>T, p.Arg1076Cys (Class 4, Insight March 2018)	LS
**33**	-		*MLH1* c.676C>T, p.Arg226* (Class 5)	-			LS
**39**	*MSH2* c.1787A>G; p.Asn596Ser (Class 3)		*MSH2* c.1787A>G; p.Asn596Ser (Class 3)	*FAN1* c.149T>G, p.Met50Arg (not reported)			LLS (MMR VUS carrier)
**42**	-		-	-			LLS
**44**	-		-	-			LLS (VUS carrier)
**45**	-		-	-			LLS
**48**	-		-	-			LLS
**53**	-		-	-			LLS
**55**	-		-	*PMS1* c.497A>C, p.Lys166Thr (not reported)			LLS (VUS carrier)
**56**	-		-	-			LLS
**57**	-		*MSH2* E8 duplication (Class 3*)	-	Aberrant splicing	***MSH2*** **E8 duplication (Class 5)**	**LS**
**58**	*MSH2* c.2045C>G; p.Thr682Ser (Class 3*)		*MSH2* c.2045C>G; p.Thr682Ser (Class 3*)	*EXO1* c.2212-1G>A (Class 3)			LLS (MMR VUS carrier)
**59**	-		-	*APC* c.1966C>G, p.Leu656Val (Class 3)			LLS (VUS carrier)
**61**	-		-	-			LLS
**62**	-		-	*MSH3* c.2732T>G, p.Leu911Trp (not reported)			LLS (VUS carrier)
**63**	*MSH2* c.2702A>T; p.Glu901Val (Class 3*)		*MSH2* c.2702A>T; p.Glu901Val (Class 3*)	-			LLS (MMR VUS carrier)
**64**	-		-	-			LLS
**65**	-		-	*MUTYH* c.1437_1439delGGA, p.Glu480del (Class 5)			LLS (monoallelic MUTYH carrier)
**66**	-		-	-			LLS
**74**		-	-	*MSH3* c.685T>C, p.Tyr229His (not reported); *MSH3* c.2732T>G, p.Leu911Trp (not reported)	MSH3 conserved expression/EMAST/in cis	*MSH3* c.685T>C, p.Tyr229His (VUS); MSH3 c.2732T>G, p.Leu911Trp (VUS)	LLS (VUS carrier)
**76**		-	-	-			LLS
**78**		-	-	-			LLS
**79**		-	-	-			LLS
**81**		-	-	*BUB1* c.2473C>T, p.Pro825Ser (not reported)			LLS (VUS carrier)
**82**		-	*MSH6* c.3150_3161dup, p.(Val1051_Ile1054dup) (Class 3*)	-	Multifactorial (>0.99)/normal splicing	***MSH6* c.3150_3161dup, p.(Val1051_Ile1054dup) (Class 5)**	**LS**
**85**		-	*PMS2* c.1320A>G, p.Pro440= (Class 3*)	*MSH3* c.3072G>C, p.Gln1024His (not reported)			LLS (MMR VUS carrier)
**87**		-	-	-			LLS
**92**		-	***MSH6* c.2219T>A, p.Leu740* (Class 5)**	-			**LS**
**93**		-	-	-			LLS
**94**		-	-	-			LLS
**95**		-	-	-			LLS
**96**		-	-	*APC* c.7514G>A, p.Arg2505Gln (Class 1,2)			LLS (VUS carrier)
**97**		-	-	-			LLS
**98**		-	*MSH2* c.2802G>A, p.Thr934Thr (Class 3)	-			LLS (MMR VUS carrier)
**B. Results obtained in 7 additional cases harboring MMR variants identified by previous Sanger sequencing**							
**35**		*MLH1* c.25C>T, p.Arg9Trp, (Class 3) *APC* c.1958+3A>G (Class 5) (Borrás et al. 2012)	-	-			**FAP (MMR VUS carrier)**
**67**		*MSH6* c.1153_1155del, p.Arg385del (Class 3 *)	-		Multifactorial (0.98)/normal splicing	***MSH6* c.1153_1155delAGG p.Arg385del (Class 4*)**	**LS**
**70**		*MSH6* c.1618_1620delCTT; p.Leu540del (Class 3 *)	-	-	Multifactorial (>0.99)/aberrant splicing at low proportion	***MSH6*** **c.1618_1620delCTT; p.Leu540del (Class 5*)**	**LS**
**72**		*MSH6* c.1450G>A; p.Glu484Lys (Class 3 *)	-	-			LLS (MMR VUS carrier)
**73**		*MSH6* c.3296T>A; p.Ile1099Asn (Class 3 *)	-	-			LLS (MMR VUS carrier)
**75**		*MSH6* c.1618_1620del; p.Leu540del (Class 3 *)	-	-	Multifactorial (>0.99)/aberrant splicing at low proportion	***MSH6*** **c.1618_1620delCTT; p.Leu540del (Class 5*)**	**LS**
**77**		*MSH6* c.3226C>T, p.Arg1076Cys (Class 3)	-	-	Insight variant classification revision	***MSH6* c.3226C>T, p.Arg1076Cys (Class 4, Insight March 2018)**	**LS**

**Table 2 cancers-12-01799-t002:** Results of the pathogenicity assessment of MMR variants of unknown significance (VUS). See Appendix A for futher details. ^Borràs et al., Hum Mut [45]; *Thompson et al., [35]; ¨Wang et al., [46]; **InSiGHT classification, March 2018. Abbreviations: NA: Not available; NP: Not performed.

Case ID	MMR Gene	MMR Variant	Predicted Protein Change	Insight Classification (2015)	ClinVar Classification	Frequency in Controls (ExAC/ESP)	RefSNP (rs)	In Silico Predictions		RNA Analyses		Multifactorial Calculations	Final Classification
								Splicing	Protein Function	cDNA Splicing Analysis	cDNA Stability Analysis (+/− Puromicin)		
**13**	*MLH1*	**c.-574T>C**	**p.?**	**Class 3**	**Not reported**	**0,000084/NR**	**rs558088820**	**NA**	NA	NP	NP	NP	Class 3
**35**		c.25C>T	p.(Arg9Trp)	Class 3	VUS (2)/+++	NR/NR	rs587779000	No changes	**Damaging**	r.25C>T^; p.Arg9Trp	NP	NP	Class 3
**39**	*MSH2*	c.1787A>G	p.(Asn596Ser)	Class 3	VUS (3) vs Bening/Likely bening (3)/+++	NR/0.0002	rs41295288	No changes	Benign	r.1787A>G; p.Asn596Ser	Biallelic expression (Sanger seq)	NP	Class 3
**57**		exon 8 duplication	p.?	Not reported	Not reported	_	_	NA	NA	r.1277_1387dup; p.Val463Glufs*11	NP	NP	**Class 5**
**58**		c.2045C>G	p.(Thr682Ser)	Not reported	Not reported	NR/NR	_	No changes	Benign	NA	NA	NP	Class 3
**63**		c.2702A>T	p.(Glu901Val)	Not reported	Not reported	NR/NR	_	No changes	**Damaging**	NA	NA	NP	Class 3
**98**		c.2802G>A	p.(Thr934=)	Class 3	VUS (2) vs Bening/Likely Bening (5)/+++	0.000/0.0001	rs150259097	No changes	NA	NP	NP	NP	Class 3
**5**	*MSH6*	c.2092C>G	p.(Gln698Glu)	Class 3	VUS (5)/+++	NR/NR	rs63750832	Unconclusive (3/5)	Benign	r.2092C>G¨; p.Gln698Glu	NP	NP	Class 3
**67**		c.1153_1155del	p.(Arg385del)	Class 3	VUS (2)/+++	NR/NR	rs267608043	No changes	**Damaging**	r.1153_1155del (NP); p.Arg385del	Non allelic imbalance (NP/1.02±0.09)	0,98	**Class 4**
**72**		c.1450G>A	p.(Glu484Lys)	Not reported	VUS (1)/+	NR/NR	_	No changes	**Damaging**	NP	NP	NP	Class 3
**70 & 75**		c.1618_1620del	p.(Leu540del)	Not reported	VUS (2) vs Pathogenic (1)/+	NR/NR	_	No changes	**Damaging**	r.[1618_1620del;1607_3172del]; p.[Leu540del;Ser536_Asp1058delinsAsn]	Destabilization (0.69±0.03/0.65±0.06)	>0,99	**Class 5**
**82**		c.3150_3161dup	p.(Val1051_Ile1054dup)	Not reported	Not reported	NR/NR	_	No changes	**Damaging**	r.3150_3161dup; p.Val1051_Ile1054dup	Non allelic imbalance(1.04±0.14/1.16±0.26)	>0,99	**Class 5**
**77**		c.3226C>T	p.(Arg1076Cys)	Class 3	Pathogenic/Likely pathogenic (6)/+++	NR/NR	rs63750617	No changes	**Damaging**	r.3226C>T*; p.Arg1076Cys	NP	NP	**Class 4****
**73**		c.3296T>A	p.(Ile1099Asn)	Not reported	Not reported	NR/NR	_	No changes	**Damaging**	NP	NP	NP	Class 3
**85**	*PMS2*	c.1320A>G	p.(Pro440=)	Not reported	VUS (1) vs Bening/Likely bening (5)/+	NR/0.0001	rs138697590	No changes	NA	NP	NP	NP	Class 3
**73**		c.3296T>A	p.(Ile1099Asn)	Not reported	Not reported	NR/NR	_	No changes	**Damaging**	NP	NP	NP	Class 3
**85**	*PMS2*	c.1320A>G	p.(Pro440=)	Not reported	VUS (1) vs Benign/likely benign (5)/+	NR/0.0001	rs138697590	No changes	NA	NP	NP	NP	Class 3

**Table 3 cancers-12-01799-t003:** Detailed multifactorial likelihood analyses of MMR VUS. Abbreviations: LR, likelihood ratio; NR, not reported; NE, not evaluable; CRC, colorectal cancer; EC, endometrial cancer; MSI-H, microsatellite instability high; MSS, microsatellite stable.

*MSH6* Variant	Frequency in Controls (ExAC/ESP)	Multifactorial Likelihood Analysis														Final Classification
		Prior Probability of PATHOGENICITY	Prior Used	Case ID	Proband (Yes/No)	Ascertainment	Cancer (Age)	MSI/IHC Status	CRC/EC LR	Tumor Characteristics LR	Bayes	Segrega-tion LR	Odds for Causality	Posterior Odds	Posterior Probability of Pathogenicity	
**c.1153_1155del**	**NR/NR**	0,134	0,5	67	Yes	clinic	CRC (53)	MSI-H & MSH6 loss	-	4,16	2,15	15,22	**63.31**	63.31	0,984	**Class 4: Likely pathogenic**
				AF1_III4	Yes	clinic	EC (59)	MSH6 loss	-		7,08					
				AF1_III1	No	clinic	CRC (59)	MSH6 loss	4,16							
**c.1618_1620del**	NR/NR	0,959	0,9	70	Yes	clinic	CRC (46)	MSI-H & MSH6 loss	-	6,54	1,85	1,85	**12,07**	108,62	0,991	**Class 5 Pathogenic**
				70	No	clinic	CRC (43)	MSI-H & MSH6 loss	6,54							
				75	Yes	clinic	CRC (45)	MSI-H & MSH6 loss	-		-					
**c.3150_3161dup**	NR/NR	0,961	0,9	82	Yes	clinic	OC (47)	MSI-H & PMS2 loss	-	30,28	-	28,75	**870,61**	7835,47	1,000	**Class 5 Pathogenic**
				AF2_II2	Yes	clinic	CRC (61)	MSI-H & MSH6 loss	-		0,99					
				AF3_III3	Yes	clinic	CRC (47)	MSH6 loss	-		29,08					
				AF3_II2	No	clinic	EC (56)	MSH6 loss	1,75							
				AF3_II2	No	clinic	CRC (75)	MSI-H & MSH6 loss	4,16							
				AF3_II11	No	clinic	CRC (68)	MSI-H	4,16							

**Table 4 cancers-12-01799-t004:** **Variants identified in non LS-associated genes and predicted pathogenic by in silico predictors**. See Appendix A for futher details. Abbreviations: NP: not performed; NA: not available; DSS=Consensus Donor Splice Site; ASS=Consensus Aceptor Splice Site; D = Damaging; PrD = Probably Damaging; PsD = Possibly Damaging; T = Tolerated. Gain or Loss of Splice sites are considered when 4 of the 5 predictors are in agreement of their calculation. Inconclusive interpretation is given when 3 of the 5 predictors predicted changes. Less than 3 similar predictions are considered as no changes. (^) Up to date only *EPCAM* 3′UTR deletions has been associated with CRC predisposition, whereas point mutations cause an autosomal recessive congenital diarrhoea (OMIM:613217) unrelated to CRC.

Case ID	Variant Calling			RefSNP (rs)	MAF	*In Silico* predictions						ClinVar Classification
						Splicing	Protein Function					
	Gene	cDNA Change	Predicted Protein Change		ExAC/ESP		SIFT (score)	Mutation Taster (p-value)	Polyphen2/HumDiv (score)	Polyphen2/HumVar (score)	Provean	
**10**	***EPCAM (^)***	**c.811G>T**	p.(Val271Phe)		NR/NR	No changes	**D (0)**	**D (1)**	**PrD (1.000)**	**PrD (0.989)**	NP	Not reported
**29**	*POLD1*	c.2275G>A	p.(Val759Ile)	rs145473716	0.002/0.001	No changes	**D (0)**	**D (1)**	**PrD (1.000)**	**PrD (0.988)**	NP	VUS (1) vs Benign/Likely benign (6)/+
**30**	*APC*	c.7936C>G	p.(Gln2646Glu)		NR/NR	No changes	**D (0.02)**	**D (1)**	**PsD (0.688)**	B (0.182)	NP	VUS (1)/+
**39**	*FAN1*	c.149T>G	p.(Met50Arg)	rs148404807	0.002/0.002	No changes	T (0.08)	**D (1)**	**PrD (0.991)**	**PsD (0.690)**	NP	Not reported
**55**	*PMS1*	c.497A>C	p.(Lys166Thr)		NR/NR	No changes	**D (0)**	**D (1)**	**PsD (0.757)**	**PsD (0.599)**	NP	Not reported
**58**	*EXO1*	c.2212-1G>A	p.Val738_Lys743del	rs4150000	0.0019/0.0028	**Loss of ASS**	NA	NA	NA	NA	NA	Lhotaa et al., 2016: r.2212_2229del; p.Val738_Lys743del
**59**	*APC*	c.1966C>G	p.(Leu656Val)	rs577466163	NR/NR	**Gain of DSS**	**D (0)**	**D (1)**	**PrD(0.999)**	**PrD (0.998)**	NP	VUS (1)/+
**62 and 74**	*MSH3*	c.2732T>G	p.(Leu911Trp)	rs41545019	0.002/0.004	No changes	**D (0)**	**D (0.999)**	**PrD (1.000)**	**PrD (0.978)**	NP	Not reported
**65**	*MUTYH*	c.1437_1439del	p.Glu480del	rs587778541	NR/0.000	No changes	NA	NA	**NA**	NA	**D (−7.78)**	Pathogenic (9)/**
**74**	*MSH3*	c.685T>C	p.(Tyr229His)		NR/NR	No changes	**D (0.01)**	**D (0.999)**	**PrD (1.000)**	**PrD (0.973)**	NP	Not reported
**81**	*BUB1*	c.2473C>T	p.(Pro825Ser)	rs748392521	NR/NR	No changes	**D (0)**	**D(1)**	**PrD (1.000)**	**PrD (0.997)**	NP	Not reported
**85**	*MSH3*	c.3072G>C	p.(Gln1024His)	rs147640909	0.000/0.000	**Loss of DSS/Inconclusive at ASS**	T (0.39)	P (0.996)	B (0.007)	B (0.013)	NP	Not reported
**96**	*APC*	c.7514G>A	p.(Arg2505Gln)	rs147549623	0.001/0.001	No changes	**D (0.04)**	**D (1)**	**PrD (1.000)**	**PrD (0.961)**	NP	Benign/Likely benign (8)/++

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
