# Peer review of "Comprehensive Constitutional Genetic and Epigenetic Characterization of Lynch-Like Individuals"

_cancers, 2020, doi:10.3390/cancers12071799_

Round 1

Reviewer 1 Report

The paper describes results of an epigenetic study using samples from 115 lynch-like syndrome individuals.  These patients had tumors with mismatch repair deficiencies which were not germ line.

The paper contains a wealth of date which will be of interest to the community but in places it is difficult to navigate and link text to figure/tables with some patient information appearing in several places.

After reading through it several times I think this may be because it jumps around using patient numbers but the authors are encouraged to consider how they might make the information on each patient better linked together. The difficulty in following the text is compounded by the order of sections which places M&M after Discussion and has the conclusions after the M&M disconnected from the Discussion. Having rechecked the instructions for authors I note they state

‘We do not have strict formatting requirements, but all manuscripts must contain the required sections: Author Information, Abstract, Keywords, Introduction, Materials & Methods, Results, Conclusions, Figures and Tables with Captions, Funding Information, Author Contributions, Conflict of Interest and other Ethics Statements’

As there is no instruction to put results before M&M I think the authors should rethink the structure of their paper not least to have the conclusion immediately after the Discussion.

Table 1 is a useful overview/summary of the cohort examined and has the merit of being in patient numerical order – the value of Figure 1 was less obvious and it was difficult to navigate as patients were not in numerical order – the authors give pedigrees for the subset of patients shown in  Figure 1 but make little use of this information – e.g. in discussing patients 33 and 92 (lines 91-105).

Line 105 – please give the patient numbers of the ’10 LSS individuals’ and also specify those of the ‘4 additional cases’.

Line 113 – case 57 is also shown in Figure 1C please cite the figure.

I think it is ill advised to have a Table with 2 sections (Table 2 A, B) – please either unify the 2 sections or renumber as separate tables.

Figure 2 sets the standard for the kind of information that is most useful to the reader – it might be useful if the authors presented examples of the novel mutations they discovered in this patient focused format e.g. the MMR truncating mutations in patients 33 and 92 for which pedigree information is buried in Figure 1.

In summary, this paper contains high value information but is very difficult to navigate.

Author Response

We thank the reviewer for his/her comments and the depth of the review.

As suggested by the reviewer, the manuscript has been reorganized in order to facilitate the comprehension. In the new version, “Patients and Methods” section has been placed after “Introduction”; and “Conclusions” are now after “Discussion” section.

As suggested by the reviewer, Figure 1 has been reorganized and pedigrees are now displayed according to the case ID numerical order.

Also, we have tried to clarify the information in line 264 (line 105 in the previous version) by adding patient ID numbers, as follows:

“In addition to the nine MMR variants of unknown significance identified in 10 LLS individuals in previous analyses (cases 35, 39, 58, 63, 67, 70, 72, 73, 75 and 77), four additional variants (MSH6 c.2092C>G, MSH6 c.3150_3161dup, PMS2 c.1320A>G and MSH2 c.2802G>A) were detected in four additional cases (cases 5, 82, 85 and 98, respectively).

Concerning the explanation about case 57, now Figure 1 is cited in line 274 (line 113 in the previous version).

As suggested, Table 2 has been splited in two separated tables (current Table 2 and Table 3).

Finally, in order to facilitate the comprehension of the text, and as suggested by reviewer 3, a schematic workflow of the study design and the obtained results are presented in Figure A1.

Reviewer 2 Report

Estela Dámaso and colleagues performed a genetic and epigenetic characterization of 115 Caucasian Lynch-like syndrome patients harboring MMR-deficient tumors and no germline MMR mutations. In this study, they performed a completed methodology approaches to assess the mutation analysis and genome-wide methylation profiling LLS selected cases. The authors identify the presence of two MMR gene truncating mutation not previously described, as well as total of 15 additional MMR variants in which some reclassified as pathogenic mutations. Additionally, in methylome analysis, the authors found one constitutional MLH1 epimutation.

The manuscript is clear, organized and well written, with suitable methodologies for the presented work. This is a positive characterization of LLS patients, with different complementary methodologies which may allow an improvement in patients stratification.

Author Response

We would like to thank reviewer 2 for his/her revision.

Reviewer 3 Report

Review of scientific report entitled “Comprehensive Constitutional Genetic and Epigenetic Characterization of Lynch-Like Individuals” by Estela Damaso et al.

The authors investigated the constitutional basis of mismatch repair (MMR) deficiency in LLS patients throughout a comprehensive (epi)genetic analysis. The authors analyzed one hundred and fifteen LLS patients harboring MMR-deficient tumors and no germline MMR mutations. Mutational analysis of 26 colorectal cancer (CRC)-associated genes was performed. Pathogenicity of MMR variants was assessed by splicing and multifactorial likelihood analyses. Genome-wide methylome analysis was performed by the Infinium HumanMethylation450K BeadChip. The authors found the presence of two MMR gene truncating mutations not previously described, using a multigene panel analysis. Moreover, of a total of 15 additional MMR variants identified, five were reclassified as pathogenic. In addition, 13 predicted deleterious variants in other CRC-predisposing genes were found in 12 probands. Methylome analysis detected one constitutional MLH1 epimutation, but no additional differentially methylated regions were identified in LLS compared to LS patients or cancer-free individuals. The authors concluded that the use of an ad-hoc designed gene panel combined with pathogenicity assessment of variants allowed the identification of deleterious MMR mutations as well as new LLS candidate causal genes. Constitutional epimutations in non-LS-associated genes are not responsible for LLS.

The work is well conducted, well written and showed very interesting data, the result of extensive work. Only two small notes:

  1. The authors should highlight more, in the discussion section, the clinical importance of the need to apply different molecular methodologies in the identification of the constitutional basis of mismatch repair (MMR) deficiency
  2. Sometimes it is difficult to follow the text therefore it would be useful to create a schematic workflow.

Author Response

We thank the reviewer for his/her comments.

  1. We have tried to highlight the clinical relevance of the need to apply different methodologies to identify the constitutional basis of MMR deficiency. A new paragraph has been added in Discussion section:

“The different molecular nature of the pathogenic MMR variants identified in our work, including single nucleotide and splicing variants, rearrangements and epimutations, highlights the need to apply a variety of experimental approaches in the search for the constitutional basis of MMR deficiency. Our comprehensive strategy has proved useful for the elucidation of the underlying molecular basis of a relevant number of suspected LS patients, with the consequent clinical impact for patients and their families.”

  1. As suggested by the reviewer and in order to facilitate the comprehension of the text, a schematic workflow of the study design and the obtained results are presented in Figure A1. Also, the manuscript has been reorganized. In the resubmitted version, “Patients and Methods” section has been placed after “Introduction”; and “Conclusions” are now after “Discussion” section.